# Antioxidant Effect of Coenzyme Q10 in the Prevention of Oxidative Stress in Arsenic-Treated CHO-K1 Cells and Possible Participation of Zinc as a Pro-Oxidant Agent

**DOI:** 10.3390/nu14163265

**Published:** 2022-08-10

**Authors:** Saulo Victor e Silva, María Celeste Gallia, Jefferson Romáryo Duarte da Luz, Adriana Augusto de Rezende, Guillermina Azucena Bongiovanni, Gabriel Araujo-Silva, Maria das Graças Almeida

**Affiliations:** 1Post-Graduation Program in Pharmaceutical Sciences, Multidisciplinary Research Laboratory, Department of Clinical and Toxicological Analysis (DACT), Health Sciences Center, Federal University of the Rio Grande do Norte (UFRN), Natal 59012570, Brazil; 2Institute of Research and Development in Process Engineering, Biotechnology and Alternative Energies (PROBIEN), National Council for Scientific and Technical Research (CONICET), School of Agricultural Sciences, Neuquén 8300, Argentina; 3Post-Graduation Program in Health Sciences, Multidisciplinary Research Laboratory, Department of Clinical and Toxicological Analysis (DACT), Health Sciences Center, UFRN, Natal 59012570, Brazil; 4Organic Chemistry and Biochemistry Laboratory, State University of Amapá (UEAP), Macapá 68900070, Brazil; 5Sciences Center, UFRN, Natal 59012570, Brazil

**Keywords:** oxidative stress, free radical, antioxidants, coenzyme Q10, Zinc

## Abstract

Oxidative stress is an imbalance between levels of reactive oxygen species (ROS) and antioxidant enzymes. Compounds with antioxidant properties, such as coenzyme Q10 (CoQ10), can reduce cellular imbalance caused by an increase in ROS. CoQ10 participates in modulating redox homeostasis due to its antioxidant activity and its preserving mitochondrial functions. Thus, the present study demonstrated the protective effects of CoQ10 against oxidative stress and cytotoxicity induced by arsenic (As). Antioxidant capacity, formation of hydroperoxides, generation of ROS, and the effect on cellular viability of CoQ10, were investigated to determine the protective effect of CoQ10 against As and pro-oxidant compounds, such as zinc. Cell viability assays showed that CoQ10 is cytoprotective under cellular stress conditions, with potent antioxidant activity, regardless of the concentration tested. Zn, when used at higher concentrations, can increase ROS and show a pro-oxidant effect causing cell damage. The cytotoxic effect observed for As, Zn, or the combination of both could be prevented by CoQ10, without any decrease in its activity at cellular levels when combined with Zn.

## 1. Introduction

Oxidative stress is well discussed in the literature and is primarily related to an imbalance between the number of reactive oxygen species (ROS) and antioxidant defenses, causing cell damage [1], decreasing antioxidant defenses, and altering the structure and function of lipids, proteins, and DNA.

ROS generation can be caused by endogenous factors, such as the activation of immune defense cells, inflammatory and infectious processes, excessive exercise, mental stress and aging, and exogenous factors, such as exposure to heavy metals, specific drugs, bad eating habits (consumption of smoked meat, fat diet), exposure to cigarette smoke, radiation, and excessive alcohol consumption [2]. Dietary compounds with antioxidant potential can reduce oxidative stress and increase the cellular defense system, preventing oxidative damage and associated diseases [2,3].

Excessive and frequent exposure to stress factors has shown an important link between oxidative stress and a wide variety of human diseases [3]. Rheumatoid arthritis [4,5], Alzheimer’s disease, Parkinson’s disease, atherosclerosis, liver damage, type 2 diabetes, kidney failure, cancer [6,7], amyotrophic lateral sclerosis (ALS) [8], cardiovascular disease, allergies and immune system disorders [9,10,11,12,13,14] are all related to oxidative stress.

Among the dietary antioxidants, coenzyme Q10 (CoQ10) can effectively inhibit the oxidation of lipids and proteins, as well as DNA damage [15]. CoQ10 is an important vitamin compound found in many living organisms, animals, and plants [16,17]. The oxidized form of CoQ10, which is most prevalent in humans and mammals, is ubiquinone, however, it is quickly converted to its reduced active form, ubiquinol (QH2:2,3-dimethoxy-5-metil-6-decaprenyl-1, 4-dihidroxybenzene), which acts as an antioxidant scavenging free radical to protect against lipid peroxidation by regenerating tocopherol, thereby preventing cell damage caused by oxidative stress [8,18]. CoQ10 also acts as a membrane stabilizer, a cofactor in ATP production, an oxidative inhibitor of proteins and DNA, and an essential component for transport in oxidative phosphorylation in the mitochondria [19]. In addition, CoQ10 has demonstrated therapeutic benefits in aging-related disorders, especially under conditions associated with increased oxidative stress [20]. Other studies have explored the importance of CoQ10 in the attenuation of other heavy metal toxicities due to its antioxidant, anti-inflammatory, and anti-apoptotic properties [21,22,23,24].

Similar to CoQ10, Zinc (Zn) is a trace element of the diet with antioxidant capacity. Zn at adequate concentrations, among other functions, also participates in the three-dimensional structure of the superoxide dismutase enzyme, which promotes the conversion of two superoxide anion radicals (O_2_^•−^) to H_2_O_2_, reducing the toxicity of ROS by converting a highly reactive species into a less reactive one, and maintaining antioxidant defense in the cell compartment [25,26,27]. In contrast, a pro-oxidant effect has also been described for Zn when its levels are either deficient or in excess, due to its numerous functions in biological systems, promoting a pro-inflammatory and pro-apoptotic effect in humans [28,29].

With these data, it was hypothesized that CoQ10 could reduce the number of damaged cells or neutralize the pro-oxidant effect of zinc during oxidative stress. The protective effect of CoQ10 could be applied to reduce or prevent possible defects in antioxidant defense and to control the oxidative stress induced by the diet. Thus, the antioxidant potential of CoQ10 was evaluated in an in vitro model using CHO-K1 cells exposed to Zn, under conditions of oxidative stress induced by As.

## 2. Materials and Methods

### 2.1. Chemical Reagents

Dulbecco’s modified essential medium (DMEM), the PeroxiDetect™ Kit, antibiotics, and reagents were purchased from Sigma-Aldrich (St. Louis, MO, USA). Fetal bovine serum (FBS) was purchased from Natocor (Córdoba, Argentina) accessed on 22 April 2019 (www.natocor.com.ar). High-performance liquid chromatography–grade solvents were purchased from Fisher Scientific Co. (Ann Arbor, MI, USA). Milli-Q water was used in all the experiments. Ubiquinol (CoQ10) and zinc (Zn) compounds in the form of zinc sulfate heptahydrate (ZnSO_4_•7H_2_O) were supplied by Farmafórmula^®^ manipulation pharmacy in Natal, RN, Brazil. CoQ10, zinc, and CoQ10 + Zn solutions were dissolved in dimethyl sulfoxide (DMSO).

### 2.2. Total Antioxidant Capacity Determination

#### 2.2.1. Antioxidant Activity Determined via Iron Reduction (FRAP)

The FRAP method assesses the reducing power of a sample based on its ability to reduce ferric iron (Fe^3+^) in its ferrous form (Fe^2+^), which is compliant with 2,4,6-tri(2-pyridyl)-s-triazine (TPTZ) with an absorbance peak at 595 nm [30]. This assay was not used to measure the antioxidant activity of Zn because it is a stable divalent cation (Zn^2+^) and does not donate electrons to Fe^3+^ because of its complete shape. Furthermore, it binds to TPTZ because it is a transition metal, forming the Zn^2+^-TPTZ complex similar to almost all transition metals [31]. For this determination, two stock solutions were prepared, 0.5 mg/mL of CoQ10 and a mixture of CoQ10 (0.5 mg/mL) and Zn (0.05 mg/mL) to make the CoQ10 + Zn association. These were diluted at 1:100, and 50 µL was added to 900 µL of FRAP solution (acetic acid-sodium acetate buffer [pH 3.4], TPTZ, and FeCl_3_, in a 10:1:1 ratio). After 30 min of reaction, the absorbance at 595 nm was determined, and these values were compared with the standard curve constructed with ascorbic acid (0–60 µg/mL). The results were expressed in milligram equivalents of ascorbic acid/mg of antioxidant (mg EAA/mg).

#### 2.2.2. Antioxidant Potential Determined via 2,2-Diphenyl-1-Picrylhydrazyl (DPPH) Radical Reduction

The colorimetric method is based on the reduction of the DPPH radical, reducing hydrazine to an absorbance of 517 nm in the presence of an antioxidant [32]. Because the reduction of the DPPH radical is controlled by the electron donation capacity of the tested molecules, this assay is not suitable for stable divalent cations (Zn^2+^). For this determination, stock solutions of CoQ10 (0.5 mg/mL) and the CoQ10 + Zn association (0.5 mg/mL and 0.05 mg/mL, respectively) were prepared. These were then diluted at 1:10, and 50 µL of each dilution was used for each determination in duplicate. Next, 50 µL of 0.625 mM ethanolic solution (50%) and 0.125 mL of 0.8 mM DPPH ethanolic solution (homogenized with a vortex in each step) were added. The kinetics of the process were immediately measured on a spectrophotometer at 517 nm in a 1 mL cuvette, and then an absorbance curve was plotted at 5, 10, 20, 30, 60, and 90 s.

According to the process kinetics results, the ability to reduce the DPPH radical was compared to that after 5 s of incubation. The following equation was used to determine the percentage of radical inhibition for each treatment over 5 s: % inhibition = [(AB − AM)/AB] × 100. where (AB) = white absorption (reaction mixture without sample), and (AM) = sample absorption.

### 2.3. CHO-K1 Cellular Studies

#### 2.3.1. In Vitro Model

Although Zn is unable to donate electrons, its redox capacity as a pro-oxidant in biological systems is well-reported [29]. Therefore, its redox activity in combination with CoQ10, or its possible protective effect against oxidative stress, was evaluated in an in vitro model. The cells used were from the Chinese hamster ovary line (CHO-K1/ATCC^®^ CCL-61), which were grown in 25 cm^2^ culture flasks containing DMEM (Sigma-Aldrich) supplemented with 10% FBS (Natocor), 100 IU/mL of sodium penicillin, and 40 µg/mL gentamicin sulfate, in a controlled atmosphere of 37 °C and 5% CO_2_. For the assays, cells were detached from the flasks by treatment with 1 mL of 0.25% trypsin (Sigma), counted in a Neubauer chamber, and plated in 96-well plates at a concentration of 1 × 10^4^ per well.

#### 2.3.2. Treatments

Ubiquinol (CoQ10), zinc (Zn, in the form of zinc sulfate heptahydrate [ZnSO_4_•7H_2_O]) and ubiquinol + zinc (CoQ10 + Zn) compounds were diluted with DMSO. The culture medium was discarded after 24 h in a CO_2_ oven at 37 °C and replaced with a new one with the following treatments: (a) CoQ_10_ in the concentrations: 0.5, 1.0 and 5.0 mg/mL; (b) Zn: 0.05, 0.1 and 0.5 mg/mL; (c) a combination of both treatments (CoQ10 + Zn): 0.55, 1.1 and 5.5 mg/mL. As was subsequently used according to previous results to determine the protective effect of treatments during oxidative stress in CHO-K_1_ cells [12,14,33].

### 2.4. Antioxidant Capacity of CoQ10 and Zn in CHO-K1 Cells

#### 2.4.1. Protective Effect against Arsenic (As) Treatments

Oxidative stress and cell death are caused by toxic elements [12,14,33,34,35,36]. Two colorimetric methods were used to detect the oxidative and cytotoxic effects of As in CHO-K1 cells as well as the protective effect of antioxidants: cell viability assay by violet crystal staining and measurement of aqueous hydroperoxides [33]. The culture medium was discarded after 24 h in a CO_2_ oven at 37 °C and replaced with new media with different concentrations of As (5, 10, and 20 ppm) for subsequent tests. DMEM was used as a negative control (without oxidative stress) and As 5 ppm was selected as a positive control (with oxidative stress).

The culture medium was replaced with a new media with the following treatments to study the effect of antioxidants: 0.5, 1.0, 5.0 mg/mL of CoQ10, 0.05, 0.1, and 0.5 mg/mL of Zn, and 0.55, 1.1 and 5.5 mg/mL of CoQ10 + Zn and the same concentrations in each treatment together with As.

#### 2.4.2. Violet Crystal Colorimetric Assay

For the method, a total of 3 × 10^3^ cells/well was used. After 2 h of treatment, the viable cells were stained with the violet crystal solution (0.5%) for 15 min and then washed with methanol (50%), and the cell morphology was evaluated through microscopy. Next, they were solubilized in 50 µL of a solution containing 0.1 M sodium citrate (pH 5.4) and 20% methanol. Cell viability was determined by morphological analysis using an optical microscope. The conical shape was considered normal (living), whereas the round shape was considered unfeasible (primarily apoptotic).

#### 2.4.3. Aqueous Hydroperoxide Determination

The decrease in the concentration of aqueous hydroperoxides was measured using the PeroxiDetect™ Kit according to the manufacturer’s instructions to determine the protective effect of the treatments in CHO-K1 cells. This procedure is based on the ability of hydroperoxides to oxidize Fe^2+^ to Fe^3+^ under acidic conditions. Oxidized iron forms an orange-colored compound called xylene, which is observed at 560 nm. Calibration curves were obtained using a solution of 100 µM H_2_O_2_. The conditions for maximum stress oxidation induced by As without significant cytotoxic effects were established by studying the time course of the formation of aqueous hydroperoxide and the influence of the As concentration. Thus, 5 ppm of As was selected from the results (Figure 1b). The cells were incubated for 48 h in a 96-well plate with cellular content, and the medium was discarded. The solutions were then treated with 5 ppm of As as the positive control, DMEM as the negative control, the three different concentrations of each treatment, and all treatments in the presence of the oxidizing agent (As). After treatment, the culture medium was removed and the cells were lysed with 1% SDS, without generating interference in the technique [12,33], incubated for 30 min, and analyzed at an absorbance of 560 nm. The data are expressed in µmol H_2_O_2_/10^4^ cells.

#### 2.4.4. Determination of Intracellular ROS

ROS generation was measured by 2′,7′-dichlorofluorescein diacetate (DCFH-DA). DCFH-DA is hydrolyzed by intracellular esterases to dichlorofluorescein (DCFH), which is trapped within the cell. This nonfluorescent molecule is then oxidized to fluorescent dichlorofluorescein (DCF) by cellular oxidants. Briefly, CHO-K1 cells were incubated in 96-well plates (1 × 10^4^ cells/well) for 24 h, and then the cells were treated with As at 5 ppm and then treated with the test compounds for 24 h. Next, the cells were incubated with a serum-free medium containing DCFH-DA (10 µM, Sigma) at 37 °C in darkness for 2 h. The fluorescence of the dye was measured using a multidetection reader (Bio-Tek Instruments Inc., Winooski, VT, USA) at excitation and emission wavelengths of 485 and 530 nm, respectively. Treated cells were visualized directly using a fluorescence microscope (Olympus, Japan) at 100× magnification [37].

### 2.5. Effect of Treatments on Cell Viability

No cytotoxic effects have been reported for CoQ10. However, it has been shown that Zn deficiency and excess Zn can cause cellular oxidative stress and that high levels of Zn^2+^ can lead to cell death from mitochondrial or lysosomal dysfunction [29]. Therefore, the MTT and Alamar Blue^®^ colorimetric methods were used to rule out the possible cytotoxic effects of the treatments against the CHO-K1 cell line.

#### 2.5.1. MTT Reduction

The MTT colorimetric method [3-(4,5-dimethylthiazol-2-yl)-2,5-diphenyltetrazolium bromide] (Sigma^®^) involves the absorption of the MTT salt by the cells, which is reduced inside the mitochondria into a product called formazan. This product accumulates inside the cell and is extracted by adding an appropriate solvent [38]. The cell suspension at a concentration of 3 × 10^4^ cell/mL was distributed in 96-well plates (100 µL per well) and incubated at 37 °C in 5% CO_2_ for adhesion. Next, 90 µL of culture medium and 10 µL of different treatment concentrations were added, and the cells were incubated again for 24 h. At the end of the incubation period, the MTT solution was added and incubated again for 4 h. The medium was carefully removed, and 100 μL of DMSO was added to solubilize the formazan crystals. The plates were then shaken for 20 min, and the absorbance of each sample was measured at 570 nm using an ELISA reader.

#### 2.5.2. Alamar Blue^®^ Reduction

The Alamar Blue^®^ colorimetric method uses a redox indicator that changes from the oxidized form (blue and nonfluorescent) to the reduced form (pink and fluorescent) in the presence of metabolically active cells. For this, the cells (3 × 10^4^ cells/well) were seeded in 96-well plates and allowed to adhere for 1 h, after which the medium was discarded, and the cells were treated at different concentrations for 24 h at 37 °C and 5% CO_2_. After treatment, Alamar Blue^®^ was added in an amount equal to 10% of the volume of the medium contained in each well, and the plate was incubated for 4 h at 37 °C and 5% CO_2_. The amount of reduced Alamar Blue^®^ was monitored by measuring the absorbance at 570 and 600 nm using ELISA [39].

### 2.6. Statistical Analysis

Data were analyzed using SPSS version 20.0 software (SPSS Inc., Chicago, IL, USA), and graphs were generated using GraphPad Prism software. Data were initially presented as descriptive measures, and the results were reported as mean ± standard deviation. Variables were initially tested for normal distribution using the Kolmogorov-Smirnov and Shapiro-Wilk tests. Parametric statistical analyses were performed using one-way ANOVA, followed by Tukey’s post-hoc test, and Pearson’s correlation was also performed. For non-parametric variables, the Kruskal-Wallis test was used, followed by Dunn’s post-test and Spearman’s correlation. The significance level was set at *p* < 0.05, with a confidence interval of 95%.

## 3. Results

CoQ10 is known in the literature for its antioxidant potential during oxidative stress in cell content. The results shown here confirm that CoQ10 has high antioxidant power and that the ability to donate electrons was not affected by a lack of Zn^2+^.

The ideal concentration of the oxidizing agent to induce cytotoxic effect was determined by evaluating cells exposed for 2 h at different concentrations of As (5, 10, and 20 ppm) and subsequent tests. At 5 ppm, changes in cell morphology but no induced apoptosis were observed, in contrast with the cells exposed to 10 and 20 ppm As (Figure 1). Therefore, this concentration was selected as a positive control (with oxidative stress).

### 3.1. Total Antioxidant Capacity of CoQ10 and Zn

#### 3.1.1. FRAP

Figure 2 shows that 1 mg of CoQ10 had a high antioxidant capacity, equivalent to 5.8 mg of ascorbic acid. The reducing potential of membrane radicals for oxidized substances is one of the properties of CoQ10. The greater the stress, the greater the cellular damage, even if CoQ10 donates electrons to O_2_^•−^—to form H_2_O_2_ inside the mitochondria. On the other hand, the combination of CoQ10 and Zn showed lower antioxidant activity compared to the action of CoQ10 alone.

#### 3.1.2. DPPH Method

The results showed that the addition of CoQ10 alone to the DPPH radical solution caused a significant reduction in the absorbance measured, presenting a greater antioxidant activity than that observed for CoQ10 + Zn (Figure 3). A solution of 0.05 mg/mL of CoQ10 inhibited 73% of the DPPH radical at 5 s, similar to the effect of a solution of 1 mg/mL of ascorbic acid, followed by CoQ10 + Zn with 52% (Figure 4). All solutions had a significant antioxidant effect, with *p* < 0.05, however, Zn reduced the effects of CoQ10.

### 3.2. Protective Effect of Treatments on Cells Exposed to As

#### Morphological State with Violet Crystal

Based on the captured images, it was observed that 5 ppm As concentration induced morphological changes on the cells when compared to the control (Figure 5a,b). This is compatible with the initial apoptosis process in which the cells lose their normal morphology, and there is swelling of the cytoplasm, organelles, and fragmentation of the DNA or cell membrane. The concentration of 5 ppm was used to assess whether there was a cytoprotective effect of the treatments used, against stress caused by exposure to As.

It was observed that the cells treated with 0.5 mg/mL of CoQ10 (Figure 5c) did not present morphological changes compared to the control, which was also found in cells treated with CoQ10 and As (Figure 5d), with CoQ10 + Zn (Figure 5g), and when combined with Zn and As (Figure 5h). This indicates that CoQ10 has a protective effect on cells, preventing changes such as oxidative damage to cells induced by As.

The cells treated with only 0.05 mg/mL of Zn (Figure 5e) also showed no changes in their morphology, except for the cells treated with Zn in the presence of As (Figure 5f), in which cells presented a more rounded shape, showing a possible cytotoxic effect at an early stage. The same change was found in the positive control (As) (Figure 5b). This indicated that Zn had no protective effect on cells damaged by As.

### 3.3. Antioxidant Potential with Reduction of Oxidative Stress by Aqueous Hydroperoxide

We observed that the higher the concentration of each treatment, the greater the reduction in hydroperoxide concentration, even when cells were exposed to the oxidizing agent (As). CoQ10 combined with Zn significantly reduced the hydroperoxide concentration (*p* < 0.05) at concentrations of 1.1 and 5.5 mg/mL, even in the presence of As (Figure 6C), and similar results were observed in cells treated with CoQ10 alone at 5.0 mg/mL (Figure 6A) and Zn at 0.5 mg/mL (Figure 6B). However, this did not affect cell viability, as demonstrated in the following sections.

### 3.4. Determination of Intracellular ROS

The generation of ROS in CHO-K1 cells, with or without pre-treatment (A, B, and C) against As-induced oxidative stress, showed that the increase in ROS production was dependent on the concentration of each pre-treatment when compared to the basal ROS production rate. The pre-treatments A and C showed a statistically significant reduction in ROS (*p* < 0.05) at their highest concentrations, with and without exposure to As, in contrast to pretreatment B, despite the protective effect (*p* < 0.05) in the reduction of ROS in cells exposed only to pre-treatment (Figure 7).

Thus, these results suggest that CoQ10 decreases As-induced ROS production for at least the first 2 h of incubation and that high Zn concentrations induce stress. The cytotoxic effect predominates with increasing Zn concentration, and therefore, possibly increasing ROS production.

### 3.5. Effects of Treatments on CHO-K1 Cell Viability

The MTT and Alamar Blue^®^ assays are based on the percentage of cell viability, in which cells treated for 24 h with CoQ10 showed a cell viability rate above 90%, even with increased concentrations. The incubation for 24 h with Zn 0.05 mg/mL did not show a cytotoxic effect, similar to that observed for violet crystal staining in cells exposed for only 2 h. However, Zn ≥ 0.1 mg/mL caused a reduction of 20% and 30% in the cell viability rate for both tests, respectively. In contrast, incubation with both antioxidants (CoQ10 + Zn) showed a similar result to CoQ10 alone (Figure 8) at concentrations of 0.5 and 1.1 mg/mL, and a reduction in the cell viability percentage for the Alamar Blue^®^ method with a statistically significant difference (*p* < 0.05) at a concentration of 5.5 mg/mL, where cell viability reached above 90%.

In contrast, we found that CoQ10 did not present a cytotoxic effect even at high concentrations. In addition, CoQ10 maintained the cell viability compromised by high concentrations of Zn.

## 4. Discussion

The use of CoQ10 is known in the literature for its antioxidant potential during oxidative stress in cell content. The results shown in the in vitro study confirmed that CoQ10 has a high antioxidant power compared to the combination with Zinc, as shown by the DPPH and FRAP tests, and that the electron-donating ability was not affected by lack of Zn^2+^. In this study, the antioxidant activity of CoQ10 was evaluated in the presence of As, Zn, and their combination as inducers of cellular oxidative stress in CHO-K1 cells. In accordance with other studies, it was observed that Zn deficiency or excess is associated with oxidative factors of nutritional status [29].

Thus, the combination between CoQ10 and other compounds with biological activity in the redox system, such as Zn, could be an alternative to reduce possible defects in the antioxidant defense and control of oxidative stress. The use of As as an inducer agent of oxidative stress and apoptosis was based on results from previous studies [14,34,36]. As also causes mitochondrial dysfunction, increased ROS, depletion of antioxidant enzymes such as GSH, as well as inactivation of proteins rich in sulfhydryl groups and lipid peroxidation, causing decreased cell viability, DNA fragmentation, altered morphology, and reduced ATP content [40,41].

At the concentrations used in this study, CoQ10, Zn, or their combination showed a protective effect by inhibiting the formation of aqueous hydroperoxides. Despite the antioxidant potential of CoQ10, one should observe the necessary concentration that can prevent and/or eliminate the excess ROS during oxidative stress, according to Wang et al. [42], when a cell undergoes stress, there is overexpression of antioxidant enzymes, reducing the condition of stress. However, factors such as insufficient concentrations of antioxidant or a significant increase in ROS causes the antioxidant activity of these enzymes to decrease, causing an increase in the oxidative state of the cell, which was not shown by the present study when CoQ10 was used.

The higher the oxidative stress, the greater the cell damage, although CoQ10 donates electrons to O_2_ to form H_2_O_2_ within the mitochondria [43]. The membrane radical reducing potential for oxidized substances is one of the properties of CoQ10. By donating a pair of electrons to O^2−^, it forms H_2_O_2_ from complex I and II to complex III. Subsequently, this H_2_O_2_ generates OH by Fenton reaction, the OH attacks membrane lipids by the action of fat-soluble vitamins binding to vitamin E, reducing the oxidative process [44].

Zinc is not redox-active and therefore does not directly interact with ROS or carbon-centered free radicals [45], such as CoQ10, since metallothionein-bound Zn will not always be able to protect all sulfhydryl groups from oxidative stress as Zn will be lost upon reaction with OH^•−^ and O2^•−^. Zn ions have a limited ability to bind the metallothionein, which is sensitive to oxidative stress situations [46,47]. In this context, exposure to heavy metals at a given concentration induces oxidative stress in the presence of As, and heavy metals such as chromium [48,49,50], lead (Pb), and mercury (Hg) are known for their toxic potential [51]. Exposure of organisms to these heavy metals at a higher level can cause overproduction of free radicals such as ROS and reactive nitrogen species [52,53], the main mechanism of heavy metal toxicity. A study [54] in rat models showed a change in cell cycle dynamics in cells exposed to As, which, combined with other compounds, presented a higher risk of toxicity.

On the other hand, due to the numerous functions of Zn in biological systems, a pro-oxidant effect has also been described when its concentration is deficient or excessive, promoting a pro-inflammatory and pro-apoptotic effect in humans, as shown in some studies [29]. When in high concentrations it becomes highly toxic [55], its excess can easily replace other metals, especially those with similar ionic radicals at the active sites of enzymes or transporters [56]. In cells, excess Zn can produce ROS and negatively influence membrane integrity and permeability [57]. As reinforced by Valko et al. [58] in their study, they observed that heavy metals are essential but can be toxic when in excess and can catalyze the generation of intracellular ROS.

The results suggest that CoQ10 and Zn decrease As-induced ROS production at least during the first 2 h of incubation and at low Zn concentrations. However, the combination of both does not enhance the individual protective effect, and upon increasing the concentration of Zn, its cytotoxic effect predominates over the protective effect, possibly increasing the production of ROS.

In our study, we observed that CoQ10 did not have a cytotoxic effect, even at high concentrations. Furthermore, CoQ10 preserved cell viability compromised by high concentrations of Zn. This was determined by means of MTT and Alamar blue^®^ cell viability assays. For this in vitro model, the Alamar Blue^®^ assay showed better sensitivity when compared to the MTT assay. When used at concentrations above 0.5 mg/mL, Zn showed a higher cytotoxic effect on CHO-K1 cells, which could suggest a lower MTT index compared to the Alamar Blue^®^ cell viability method. Similar results were found in studies that evaluated drugs with antioxidant potential [59]. A study also observed that the Alamar Blue^®^ assay was more sensitive than the MTT assay for different compounds evaluated and did not detect cytotoxicity in some compounds after 24 h of exposure [60].

The results reported here suggest that CoQ10 and Zn decrease As-induced ROS production at least during the first 2 h of incubation and at low Zn concentrations. However, the combination of both does not enhance the individual protective effect, and by increasing the concentration of Zn, its cytotoxic effect predominates over the protective effect, possibly increasing the production of ROS. Many antioxidants decrease intracellular ROS levels through inhibition of complex I of the mitochondrial respiratory chain [61]. Some antioxidants, in addition to acting as enzyme cofactors such as CoQ10 [19], also rescue As toxicity and are classified as ROS scavengers, oxidative enzyme inhibitors, metal chelators, and enzyme cofactors such as zinc [62].

No association between CoQ10 and Zn is suggested, but, if associated, low concentrations of Zn should be used in terms of treatments to improve antioxidant defenses, and As to induce stress. As showed by Kumar and Sharma [51], a significant reduction in GSH level was observed in the brain of the As-treated group compared to that in the control group, whereas the GSH level was higher in the CoQ10-treated group for protected groups, compared to that in the As-exposed group.

Excessive ROS generation plays an important role in the molecular mechanism of As-induced toxicity and some diseases. Antioxidants have been widely studied as therapeutic agents to combat As-induced toxicity. The CoQ10 donates a pair of electrons to O_2_^•−^ and forms H_2_O_2_ from complexes I and II to complex III. This H_2_O_2_ generates OH^•^ by Fenton’s reaction, leading to OH^•^ attacking membrane lipids by the action of fat-soluble vitamins binding to vitamin E, and therefore reducing the oxidative process.

However, in Figure 8B it is shown that Zinc induces cell death but, at the same time, that Zinc does not induce oxidative stress without Arsenic (Figure 7B). Although zinc protects cells against oxidative damage, it acts in the stabilization of membranes, inhibits the enzyme nicotinamide adenine dinucleotide phosphate oxidase (NADPH-Oxidase), a pro-oxidant enzyme [45,63,64] which results in decreased generation of ROS.

Our study observed that the antioxidant activity of the CoQ10, the same at higher concentrations, prevents the increase of ROS without causing cytotoxic effect. This is unlike Zn, which, despite its properties as an antioxidant, when used at higher concentrations, generated cellular toxicity with a possible pro-oxidant effect, which is prevented by the antioxidant activity of CoQ10, which showed no increase in its activity at the cellular level when combined with Zn.

## 5. Conclusions

The in vitro oxidative stress model using CHO-K1 cells showed that the antioxidant activity of CoQ10 can prevent cytotoxic effects and oxidative damage caused by As and Zn at high concentrations. Moreover, the association of CoQ10 with Zn may reduce the antioxidant action of CoQ10. Therefore, it is necessary to conduct further studies on the application of CoQ10 to reduce toxic cellular effects caused by the exposure of specific oxidizing agents to oxidative stress.

## Figures and Tables

**Figure 1 nutrients-14-03265-f001:**
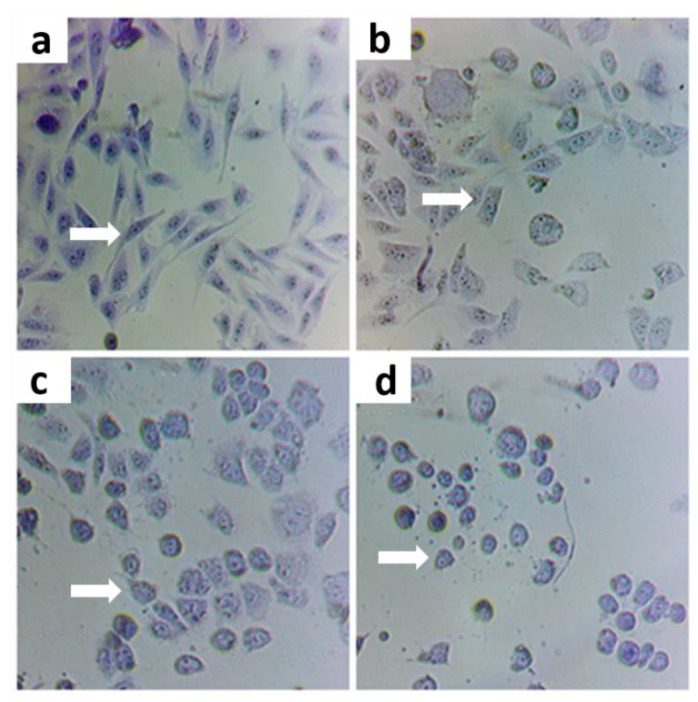
Effect of As as Negative control (**a**) with positive controls of As 5 ppm (**b**), As 10 ppm (**c**), and 20 ppm (**d**) concentration on CHO-K1 cell morphology. Image taken with a camera under a 200× optical microscope.

**Figure 2 nutrients-14-03265-f002:**
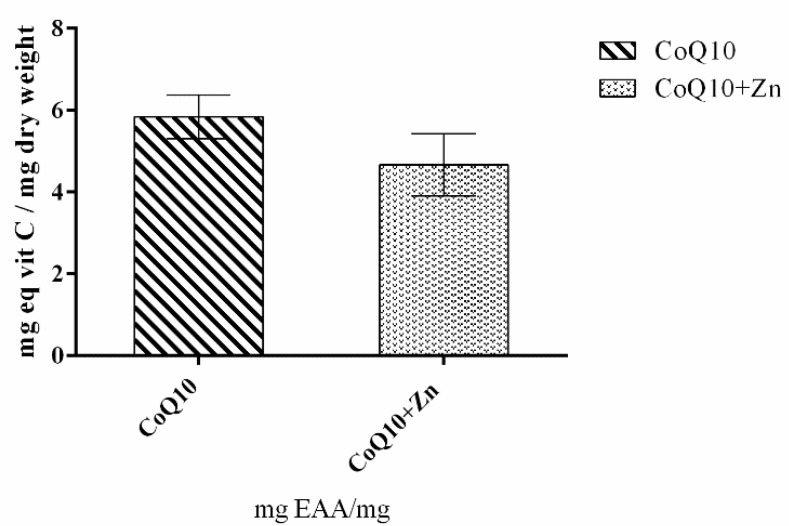
Antioxidant activity of CoQ10, and CoQ10 + Zn determined using the ferric reduction power assay (FRAP). The extracts were analyzed, and the values were compared with the standard curve constructed with ascorbic acid (0–60 µg/mL).

**Figure 3 nutrients-14-03265-f003:**
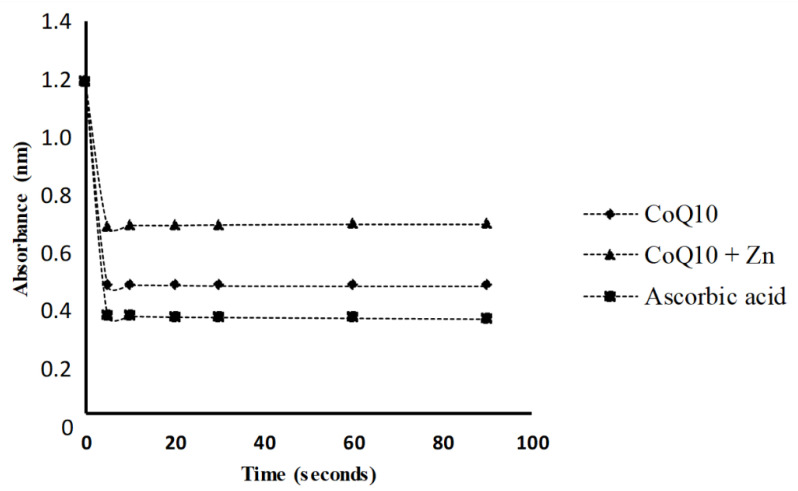
DPPH radical reduction kinetics in the presence of CoQ10 (0.05 mg/mL) and CoQ10 + Zn (0.05 + 0.005 mg/mL, respectively), expressed from a standard ascorbic acid starting point (1 mg/mL).

**Figure 4 nutrients-14-03265-f004:**
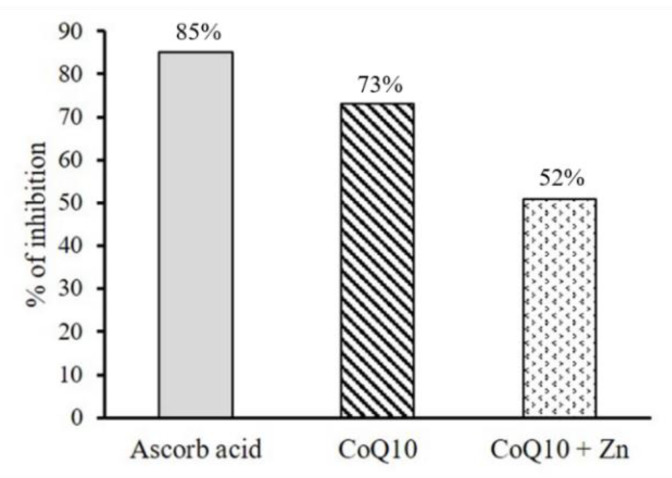
Percentage of inhibition of DPPH radical in 5 s by the antioxidant activity of CoQ10 (0.5 mg/mL) and CoQ10 + Zn (0.5 + 0.05 mg/mL, respectively) compared to the ascorbic acid standard (1 mg/mL).

**Figure 5 nutrients-14-03265-f005:**
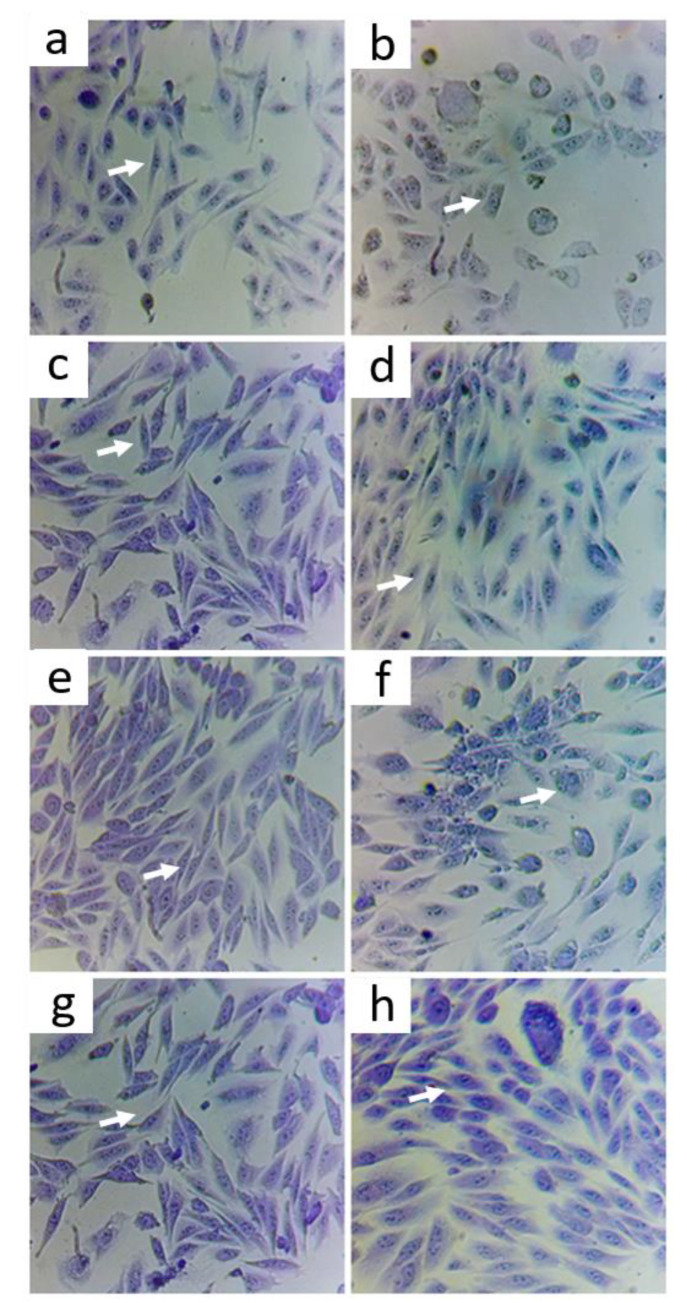
Violet crystal staining to determine viability in cells treated for 2 h with 0.5 mg/mL CoQ10, 0.05 mg/mL Zn, and 0.5 mg/mL CoQ10 + 0.05 mg/mL Zn compared with that seen with the same treatments in the presence of As at 5 ppm. (**a**) control, (**b**) As, (**c**) CoQ10, (**d**) CoQ10 + As, (**e**) Zn, (**f**) Zn + As, (**g**) CoQ10 + Zn, (**h**) CoQ10 + Zn + As. Image taken with a camera under a 200× optical microscope.

**Figure 6 nutrients-14-03265-f006:**
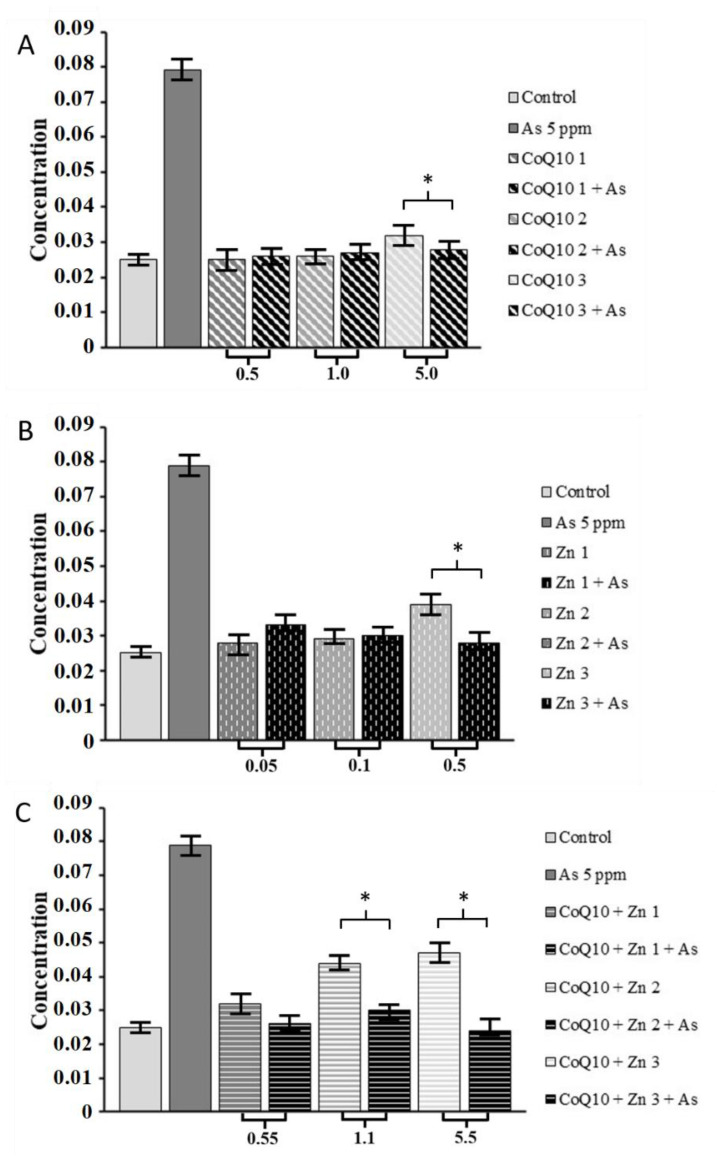
Determination of concentration of aqueous hydroperoxide in the presence of As at 5 ppm at concentrations different. (**A**) of CoQ10 1 at 0.5 mg/mL, CoQ10 2 at 1.0 mg/mL and CoQ10 3 at 5.0 mg/mL; (**B**) of Zn 1 at 0.05 mg/mL, Zn 2 at 0.1 mg/mL and Zn 3 at 0.5 mg/mL; and (**C**) at CoQ10 + Zn 1 at 0.5 + 0.05 mg/mL, CoQ10 + Zn 2 at 1.0 + 0.1 mg/mL and CoQ10 + Zn 3 at 5.0 + 0.5 mg/mL, respectively, and each antioxidant treatment combined with As at 5 ppm. Statistical analysis was performed by one-way ANOVA test, followed by Tukey’s post-hoc test. * (*p* < 0.05).

**Figure 7 nutrients-14-03265-f007:**
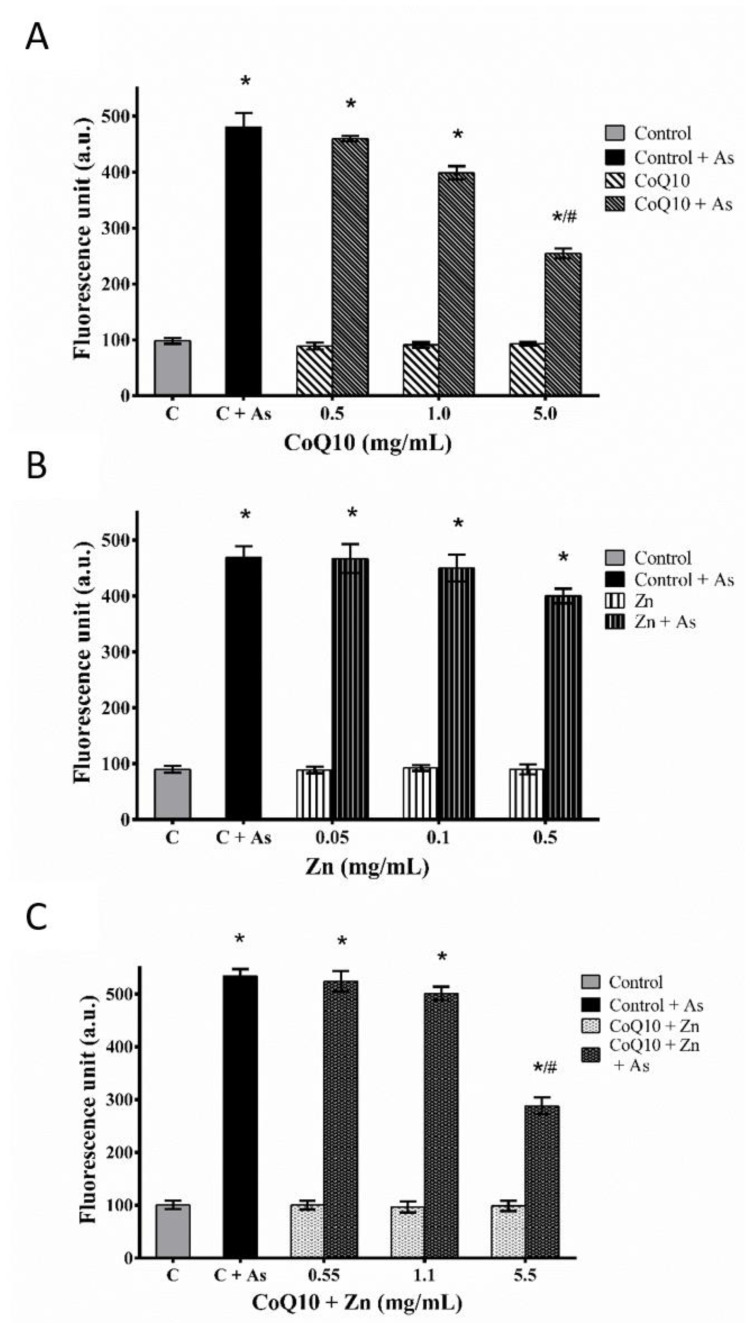
Protective effect of CoQ10 (**A**), Zn (**B**) and CoQ10 + Zn (**C**) pretreatment against As-induced oxidative stress in CHO-K1 cells. The CHO-K1 cells were exposed to 5 ppm of As for 24 h with or without test compounds pretreatment. ROS generation was measured by the DCFH-DA method using a multidetection reader and fluorescence microscope at 100× magnification. Statistical analysis was performed by one-way ANOVA test, followed by Tukey’s post-hoc test. The results are expressed as the means ± SEM of three independent experiments. * (*p* < 0.05) versus the same compound group without As, # (*p* < 0.05) versus between groups with As.

**Figure 8 nutrients-14-03265-f008:**
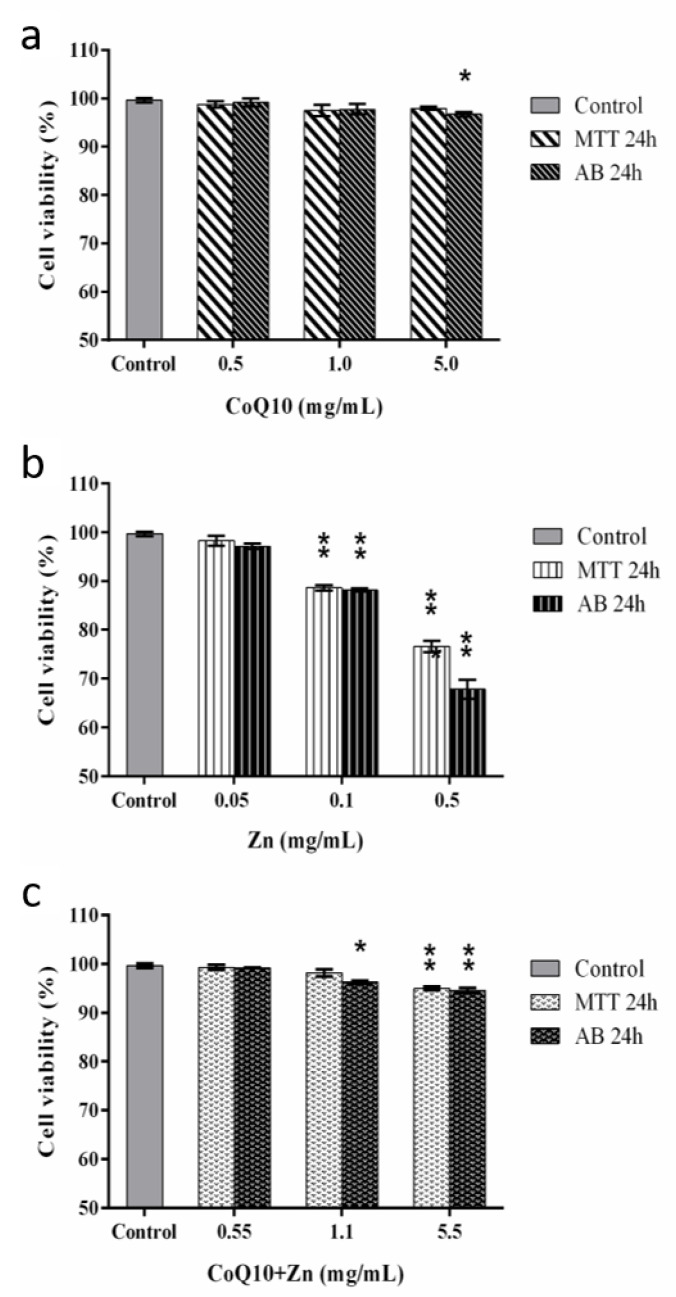
Cell viability by the MTT and Alamar Blue^®^ methods after 24 h of exposure in CHOK-1 cells treated with CoQ10 (**a**), Zn (**b**) and CoQ10 + Zn (**c**) in different concentrations. Statistical analysis was performed by one-way ANOVA test, followed by Tukey’s post-hoc test. The *p*-values are represented by * (*p* = 0.02–0.009) and ** (*p* ≤ 0.000).

## Data Availability

Not applicable.

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
