# Peer review of "Antioxidant Effect of Coenzyme Q10 in the Prevention of Oxidative Stress in Arsenic-Treated CHO-K1 Cells and Possible Participation of Zinc as a Pro-Oxidant Agent"

_nutrients, 2022, doi:10.3390/nu14163265_

Round 1

Reviewer 1 Report

Victor e Silva et al,. proposes that CoQ10 can neutralize cytotoxic caused by excess Zinc. However, the manuscript can be further improved:

1. May the authors repeat the survival experiments in figure.8 in other cell lines to strengthen the the conclusion?

2. CoQ10 mainly functions as a diffusible electron carrier in the mitochondrial respiratory chain. Zinc is also important cofactor for SOD activity in mitochondria. The authors should measure mitochondrial oxidative stress (mitoSox), membrane potential (TMRM), and OxPhos activity (ATP production) under oxidative stress. It'll may better explain how CoQ10 regulate cell survival by excess Zinc. 

3. Besides, the author should explain how Zinc induce cell death (figure. 8b), but at the same time showing Zinc doesn't induce oxidative stress without Arsenic (figure. 7B). Does excess Zinc has negative effects on other cellular functions?

Reviewer 2 Report

This manuscript details a cytoprotective effect of CoQ10 in the face of arsenic challenge in an in vitro model. The conclusions match the data presented, but some modifications would be beneficial (please see specific comments below). 

Line 52: Saying “antioxidant enzymes” is a bit misleading here, as antioxidants are comprised of enzymatic and nonenzymatic substances. “Antioxidant defenses” would be more appropriate.

Line 65: Suggest moving this sentence to the paragraph above.

Line 93: Please be more specific when you say “alternative.” As in, it could be an alternative to what?

Line 153/Treatments: Did you include a vehicle/DMSO control in your experiment? DMSO can have cytotoxic effects on cells and this would have been an important control to include.

Line 205: Please make the 4 a superscript

Figure 1: Suggest moving to below the text ending at line 265

3.1.1: Please also describe CoQ10 + Zn treatment results

Lines 269-274: This information is better suited in the discussion.

Figure 3: Are the error bars missing in this figure?

Line 322: Please add figure number (e.g., say “Figure 6C”)

Figure 6: Please provide more detail in the caption or change the figure legend. For instance, it is not defined what “CoQ10 1” vs “CoQ10 2” means. The reader can gather it by looking below the x-axis and seeing that there were different concentrations, but as it stands right now, the figure is confusing to follow.

Figure 7A&B: Some of your legend has been cut off. Please revise.

Lines 376-383: These lines are better suited for the discussion.

Line 413: Please define MT.

Discussion: Currently, the discussion reads primarily as a list of facts from other studies, without much relation to the specific data provided in this manuscript. The discussion would benefit from more direct connections made between your data and the literature that you have cited. For instance, you discuss ROS production starting at line 405; however, you do not discuss the implications of your results in this context.

Round 2

Reviewer 1 Report

The manuscript can't be published in current format until the authors are able to perform minimum experiments to response to my questions.

Author Response

Por favor, verifique o anexo.

Reviewer 2 Report

The revised manuscript demonstrates several improvements that will benefit the reader. There is only one change I would recommend, which is to include a brief statement in the results saying you performed DMSO control but no detrimental effects were seen/it had no effect on your results. Without such a statement, the reader may question whether or not this control was included or they may doubt your results.

Author Response

This manuscript is a resubmission of an earlier submission. The following is a list of the peer review reports and author responses from that submission.

Round 1

Reviewer 1 Report

The manuscript submitted by Dr. Maria Almeida and co-workers focuses on the antioxidant effect of CoQ10 when there are arsenic and zinc stressors. Data are presented and analyzed from two perspectives, ex-vivo analysis using FRAP and DPPH methods, and in vitro cell line analysis. There are many analyzes on the antioxidant effects of some chemical elements, such as cadmium and other metals or metalloids, and comparisons with the beneficial effects that CoQ10 has, but, indeed, there are no detailed data regarding the interaction with arsenic and, surprisingly, with zinc, although cytotoxic or beneficial effects are studied in both components. Therefore, the novelty of the subject is somewhat assured.

Overall, the manuscript is systematically, clear and concise written, provides the necessary information to understand the motivation and methods described, but the description / interpretation of the data is somewhat deficient. Consequently, here are some suggestions / remarks that should be addressed to the authors to revise and improve their work before an eventual acceptance for publishing in Nutrients journal:

  1. At FRAP and DPPH method, only one concentration of CoQ10 is analyzed, 0.5 mg / ml CoQ and the mixture of 0.5 mg / ml CoQ10 + 0.05 mg / ml Zn. The authors do not motivate why they chose only one case of concentration for their study and, regardless of motivation, in order to understand the following results, several concentrations must be analyzed. For example, the authors could analyze at least 4 concentrations of CoQ10, compare them and choose one of them (0.5 mg / ml or something else) to refer to zinc. Zinc reporting should be done with a fixed concentration of CoQ10 (chosen from the first graph) and variable concentrations of zinc (at least 4 concentrations).

This approach could help to better understand how the CoQ10 - Zn mixture works. Please describe the results by comparing the values and the trend obtained.

  1. In section 3.2.1. The text is confusingly written, it is relatively difficult to understand the effect of the dream CoQ10 + Zn. Please express more clearly the comparison between fig5G and 5H. Because the results in this section are not quantifiable by numbers, it is more difficult to work with different concentrations of components, so fixed concentrations can only be kept for mnorphological description, as has already been done.

  1. In section 3.3, please present the results in at least the same way as in sections 3.4 and 3.5, using several concentrations of CoQ10, Zn and CoQ10 + Zn (together with As). Describe the results in more detail where similar effects are observed or where there are large differences.

Overall, the authors need to comment on the results in more detail (simple comparisons or interpretations) or discuss them more in the discussion (where commenting on the results is only very brief).

Reviewer 2 Report

The authors of the manuscript “Does Coenzyme Q10 can prevent……….induced? did try to show if CoQ10 does protect against oxidative stress induced by Zinc and Arsenic but the authors failed by a huge margin.  The authors tried to use in vitro model in CHO-K1 cell line the toxicity and the antioxidant activity of CoQ10 to prevent the oxidative stress and damage caused by As and Zn.   It seems the authors had this tit bit data from here and there and put them without the fundamental understanding of oxidative stress and where the CoQ10 acts or functions. But the authors have made some fundamental mistakes.

  1. The authors in the manuscript are taking about CoQ10 and there is no data on the mitochondrial specific ROS in the manuscript. The authors also fail in showing how the CoQ10 might act on the antioxidant system (antioxidant enzymes) to prevent against oxidative stress.
  2. How did the authors know that CoQ10 that was administered reach which part of the cell. If CoQ10 reaches the cytosol then due to the acidic nature the CoQ10 will get oxidized and how will it function. Does CoQ10 have any action or affect in the cytosol.
  3. Why did the authors use so less amount (5ppm) of As to induce oxidative stress and not use more As to produce more oxidative stress? Why the authors used As and not Cr (VI) w
  4. Which would have produced more oxidative stress.
  5. In the whole manuscript the authors bring in the major player in the oxidative stress, GSH in the discussion part and in there is no mention of it in the introduction or the authors never showed any data with GSH.
  6. CoQ10 acts in the electron transport chain and hence the authors should have shown some data on that part which the authors very conveniently never showed any data.
  7. Also why the authors pretreated the cells with CoQ10 and Zn. Could not understand the why the authors in the title show oxidative stress induced by Zn and As. As of for As its fine but why Zn was bit confusing.
  8. In cell viability study with Zn at 0.1 mg/ml there is significant loss of cell viability but when you check the ROS using DCFDA there is no change in the ROS. The authors in the ROS study used pretreatment.  The authors are themselves confused.
  9. The whole manuscript would great if the authors showed that CoQ10 did affect any other antioxidant enzymes and that’s how CoQ10 exerts the antioxidant status which the authors have not even tried.
  10. The statistical analysis used in the manuscript is also not very informative.

Round 2

Reviewer 1 Report

In my opinion, the authors have made sufficient efforts to improve the work and the results are properly correlated, having an experimental flow that provides suitable data for publication.